# Depth-Adaptive Transformer

**Maha Elbayad**[*]
Univ. Grenoble Alpes

**Jiatao Gu, Edouard Grave, Michael Auli**
Facebook AI Research

## Abstract

State of the art sequence-to-sequence models for large scale tasks perform a fixed number of computations for each input sequence regardless of whether it is easy or hard to process. In this paper, we train Transformer models which can make output predictions at different stages of the network and we investigate different ways to predict how much computation is required for a particular sequence. Unlike dynamic computation in Universal Transformers, which applies the same set of layers iteratively, we apply different layers at every step to adjust both the amount of computation as well as the model capacity. On IWSLT German-English translation our approach matches the accuracy of a well tuned baseline Transformer while using less than a quarter of the decoder layers.

## 1 Introduction

The size of modern neural sequence models (Gehring et al., 2017; Vaswani et al., 2017; Devlin et al., 2019) can amount to billions of parameters (Radford et al., 2019). For example, the winning entry of the WMT'19 news machine translation task in English-German used an ensemble totaling two billion parameters (Ng et al., 2019). While large models are required to do better on hard examples, small models are likely to perform as well on easy ones, *e.g.*, the aforementioned ensemble is probably not required to translate a short phrase such as "Thank you". However, current models apply the same amount of computation regardless of whether the input is easy or hard.

In this paper, we propose Transformers which adapt the number of layers to each input in order to achieve a good speed-accuracy trade off at inference time. We extend Graves (2016; ACT) who introduced dynamic computation to recurrent neural networks in several ways: we apply different layers at each stage, we investigate a range of designs and training targets for the halting module and we explicitly supervise through simple oracles to achieve good performance on large-scale tasks.

Universal Transformers (UT) rely on ACT for dynamic computation and repeatedly apply the same layer (Dehghani et al., 2018). Our work considers a variety of mechanisms to estimate the network depth and applies a different layer at each step. Moreover, Dehghani et al. (2018) fix the number of steps for large-scale machine translation whereas we vary the number of steps to demonstrate substantial improvements in speed at no loss in accuracy. UT uses a layer which contains as many weights as an entire standard Transformer and this layer is applied several times which impacts speed. Our approach does not increase the size of individual layers. We also extend the resource efficient object classification work of Huang et al. (2017) and Bolukbasi et al. (2017) to structured prediction where dynamic computation decisions impact future computation. Related work from computer vision includes Teerapittayanon et al. (2016); Figurnov et al. (2017) and Wang et al. (2018) who explored the idea of dynamic routing either by exiting early or by skipping layers.

We encode the input sequence using a standard Transformer encoder to generate the output sequence with a varying amount of computation in the decoder network. Dynamic computation poses a challenge for self-attention because omitted layers in prior time-steps may be required in the future. We experiment with two approaches to address this and show that a simple approach works well (§2). Next, we investigate different mechanisms to control the amount of computation in the decoder network, either for the entire sequence or on a per-token basis. This includes multinomial and binomial classifiers supervised by the model likelihood or whether the argmax is already correct as well as simply thresholding the model score (§3). Experiments on IWSLT14 German-English

---

[*]Work done during an internship at Facebook AI Research.

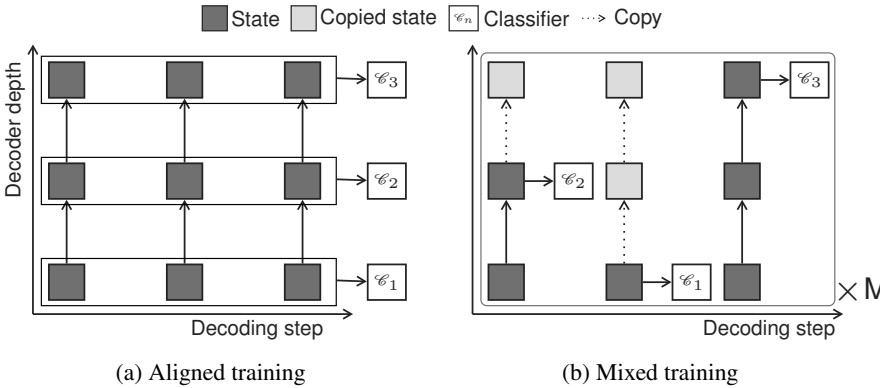

Figure 1: Training regimes for decoder networks able to emit outputs at any layer. Aligned training optimizes all output classifiers $\mathscr{C}_n$ simultaneously assuming all previous hidden states for the current layer are available. Mixed training samples $M$ paths of random exits at which the model is assumed to have exited; missing previous hidden states are copied from below.

translation (Cettolo et al., 2014) as well as WMT'14 English-French translation show that we can match the performance of well tuned baseline models at up to 76% less computation (§4).

## 2 ANYTIME STRUCTURED PREDICTION

We first present a model that can make predictions at different layers. This is known as *anytime prediction* for computer vision models (Huang et al., 2017) and we extend it to structured prediction.

### 2.1 TRANSFORMER WITH MULTIPLE OUTPUT CLASSIFIERS

We base our approach on the Transformer sequence-to-sequence model (Vaswani et al., 2017). Both encoder and decoder networks contain $N$ stacked blocks where each has several sub-blocks surrounded by residual skip-connections. The first sub-block is a multi-head dot-product self-attention and the second a position-wise fully connected feed-forward network. For the decoder, there is an additional sub-block after the self-attention to add source context via another multi-head attention.

Given a pair of source-target sequences $(\boldsymbol{x}, \boldsymbol{y})$, $\boldsymbol{x}$ is processed with the encoder to give representations $\boldsymbol{s} = (s_1, \ldots, s_{|\boldsymbol{x}|})$. Next, the decoder generates $\boldsymbol{y}$ step-by-step. For every new token $\boldsymbol{y}_t$ input to the decoder at time $t$, the $N$ decoder blocks process it to yield hidden states $(h_t^n)_{1 \leq n \leq N}$:

$$h_t^0 = \text{embed}(y_t), \quad h_t^n = \text{block}_n(h_{\leq t}^{n-1}, \boldsymbol{s}), \tag{1}$$

where $\text{block}_n$ is the mapping associated with the $n^{\text{th}}$ block and $\text{embed}$ is a lookup table.

The output distribution for predicting the next token is computed by feeding the activations of the last decoder layer $h_t^N$ into a softmax normalized output classifier $W$:

$$p(y_{t+1}|h_t^N) = \text{softmax}(W h_t^N) \tag{2}$$

Standard Transformers have a single output classifier attached to the top of the decoder network. However, for dynamic computation we need to be able to make predictions at different stages of the network. To achieve this, we attach output classifiers $\mathscr{C}_n$ parameterized by $W_n$ to the output $h_t^n$ of each of the $N$ decoder blocks:

$$\forall n, \ p(y_{t+1}|h_t^n) = \text{softmax}(W_n h_t^n) \tag{3}$$

The classifiers can be parameterized independently or we can share the weights across the $N$ blocks.

### 2.2 TRAINING MULTIPLE OUTPUT CLASSIFIERS

Dynamic computation enables the model to use any of the $N$ exit classifiers instead of just the final one. Some of our models can choose a different output classifier at each time-step which results in an exponential number of possible output classifier combinations in the sequence length.

We consider two possible ways to train the decoder network (Figure 1). *Aligned training* optimizes all classifiers simultaneously and assumes all previous hidden states required by the self-attention are available. However, at test time this is often not the case when we choose a different exit for every token which leads to misaligned states. Instead, *mixed training* samples several sequences of exits for a given sentence and exposes the model to hidden states from different layers.

Generally, for a given output sequence $\boldsymbol{y}$, we have a sequence of chosen exits $(n_1, \ldots, n_{|\boldsymbol{y}|})$ and we denote the block at which we exit at time $t$ as $n_t$.

### 2.2.1 ALIGNED TRAINING

Aligned training assumes all hidden states $h_1^{n-1}, \ldots, h_t^{n-1}$ are available in order to compute self-attention and it optimizes $N$ loss terms, one for each exit (Figure 1a):

$$\mathrm{LL}_t^n = \log p(y_t | h_{t-1}^n), \quad \mathrm{LL}^n = \sum_{t=1}^{|\boldsymbol{y}|} \mathrm{LL}_t^n, \quad \mathcal{L}_{dec}(\boldsymbol{x}, \boldsymbol{y}) = -\frac{1}{\sum_n \omega_n} \sum_{n=1}^N \omega_n \, \mathrm{LL}^n. \quad (4)$$

The compound loss $\mathcal{L}_{dec}(\boldsymbol{x}, \boldsymbol{y})$ is a weighted average of $N$ terms w.r.t. to $(\omega_1, \ldots \omega_N)$. We found that uniform weights achieve better BLEU compared to other weighing schemes (*c.f.* Appendix A). At inference time, not all time-steps will have hidden states for the current layer since the model exited early. In this case, we simply *copy* the last computed state to all upper layers, similar to mixed training (§2.2.2). However, we do apply layer-specific key and value projections to the copied state.

### 2.2.2 MIXED TRAINING

Aligned training assumes that all hidden states of the previous time-steps are available but this assumption is unrealistic since an early exit may have been chosen previously. This creates a mismatch between training and testing. Mixed training reduces the mismatch by training the model to use hidden states from different blocks of previous time-steps for self-attention. We sample $M$ different exit sequences $(n_1^{(m)}, \ldots n_{|\boldsymbol{y}|}^{(m)})_{1 \le m \le M}$ and evaluate the following loss:

$$\mathrm{LL}(n_1, \ldots, n_{|\boldsymbol{y}|}) = \sum_{t=1}^{|\boldsymbol{y}|} \log p(y_t | h_{t-1}^{n_t}), \quad \mathcal{L}_{dec}(\boldsymbol{x}, \boldsymbol{y}) = -\frac{1}{M} \sum_{m=1}^M \mathrm{LL}(n_1^{(m)}, \ldots, n_{|\boldsymbol{y}|}^{(m)}). \quad (5)$$

When $n_t < N$, we copy the last evaluated hidden state $h_t^n$ to the subsequent layers so that the self-attention of future time steps can function as usual (see Figure 1b).

## 3 ADAPTIVE DEPTH ESTIMATION

We present a variety of mechanisms to predict the decoder block at which the model will stop and output the next token, or when it should *exit* to achieve a good speed-accuracy trade-off. We consider two approaches: *sequence-specific depth* decodes all output tokens using the same block (§3.1) while *token-specific depth* determines a separate exit for each individual token (§3.2).

We model the distribution of exiting at time-step $t$ with a parametric distribution $q_t$ where $q_t(n)$ is the probability of computing $\mathrm{block}_1, \ldots, \mathrm{block}_n$ and then emitting a prediction with $\mathscr{C}_n$. The parameters of $q_t$ are optimized to match an oracle distribution $q_t^*$ with cross-entropy:

$$\mathcal{L}_{\mathrm{exit}}(\boldsymbol{x}, \boldsymbol{y}) = \sum_t H(q_t^*(\boldsymbol{x}, \boldsymbol{y}), q_t(\boldsymbol{x})) \quad (6)$$

The exit loss ($\mathcal{L}_{\mathrm{exit}}$) is back-propagated to the encoder-decoder parameters. We simultaneously optimize the decoding loss (Eq. (4)) and the exit loss (Eq. (6)) balanced by a hyper-parameter $\alpha$ to ensure that the model maintains good generation accuracy. The final loss takes the form:

$$\mathcal{L}(\boldsymbol{x}, \boldsymbol{y}) = \mathcal{L}_{dec}(\boldsymbol{x}, \boldsymbol{y}) + \alpha \mathcal{L}_{\mathrm{exit}}(\boldsymbol{x}, \boldsymbol{y}), \quad (7)$$

In the following we describe for each approach how the exit distribution $q_t$ is modeled (illustrated in Figure 2) and how the oracle distribution $q_t^*$ is inferred.

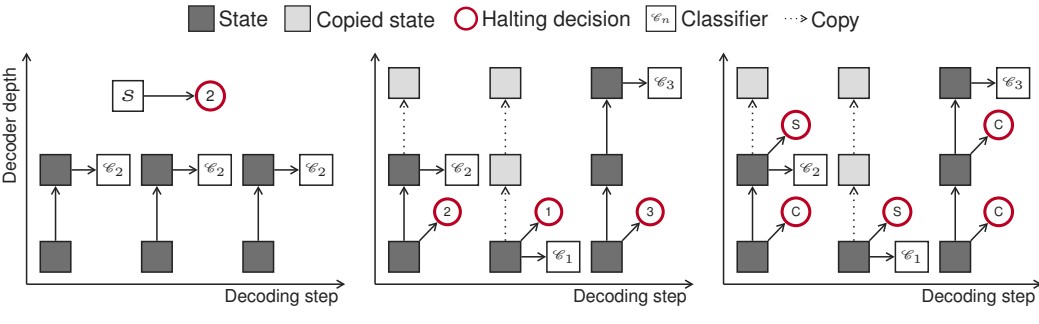

(a) Sequence-specific depth     (b) Token-specific - Multinomial     (c) Token-specific - Geometric-like

Figure 2: Variants of the adaptive depth prediction classifiers. Sequence-specific depth uses a multinomial classifier to choose an exit for the entire output sequence based on the encoder output $s$ (2a). It then outputs a token at this depth with classifier $\mathscr{C}_n$. The token-specific multinomial classifier determines the exit after the first block and proceeds up to the predicted depth before outputting the next token (2b). The token geometric-like classifier (2c) makes a binary decision after every block to dictate whether to continue (C) to the next block or to stop (S) and emit an output distribution.

### 3.1 SEQUENCE-SPECIFIC DEPTH:

For sequence-specific depth, the exit distribution $q$ and the oracle distribution $q^*$ are independent of the time-step so we drop subscript $t$. We condition the exit on the source sequence by feeding the average $s$ of the encoder outputs to a multinomial classifier:

$$s = \frac{1}{|\boldsymbol{x}|} \sum_t s_t, \quad q(n|\boldsymbol{x}) = \text{softmax}(W_h s + b_h) \in \mathbb{R}^N, \tag{8}$$

where $W_h$ and $b_h$ are the weights and biases of the halting mechanism. We consider two oracles to determine which of the $N$ blocks should be chosen. The first is based on the sequence likelihood and the second looks at an aggregate of the correctly predicted tokens at each block.

**Likelihood-based:** This oracle is based on the likelihood of the entire sequence after each block and we optimize it with the Dirac delta centered around the exit with the highest sequence likelihood.

$$q^*(\boldsymbol{x}, \boldsymbol{y}) = \delta(\arg\max_n \text{LL}^n).$$

We add a regularization term to encourage lower exits that achieve good likelihood:

$$q^*(\boldsymbol{x}, \boldsymbol{y}) = \delta(\arg\max_n \text{LL}^n - \lambda n). \tag{9}$$

**Correctness-based:** Likelihood ignores whether the model already assigns the highest score to the correct target. Instead, this oracle chooses the lowest block that assigns the largest score to the correct prediction. For each block, we count the number of correctly predicted tokens over the sequence and choose the block with the most number of correct tokens. A regularization term controls the trade-off between speed and accuracy.

$$C^n = \#\{t \,|\, y_t = \arg\max_y p(y|h_{t-1}^n)\}, \quad q^*(\boldsymbol{x}, \boldsymbol{y}) = \delta(\arg\max_n C^n - \lambda n). \tag{10}$$

Oracles based on test metrics such as BLEU are feasible but expensive to compute since we would need to decode every training sentence $N$ times. We leave this for future work.

### 3.2 TOKEN-SPECIFIC DEPTH:

The token-specific approach can choose a different exit at every time-step. We consider two options for the exit distribution $q_t$ at time-step t: a multinomial with a classifier conditioned on the first decoder hidden state $h_t^1$ and a geometric-like where an exit probability $\chi_t^n$ is estimated after each block based on the activations of the current block $h_t^n$.

**Multinomial $q_t$:**

$$q_t(n|\boldsymbol{x}, \boldsymbol{y}_{<t}) = \text{softmax}(W_h h_t^1 + b_h), \tag{11}$$

The most probable exit $\arg\max q_t(n|\boldsymbol{x}, \boldsymbol{y}_{<t})$ is selected at inference.

**Geometric-like $q_t$:**

$$\forall n \in [1..N-1], \ \chi_t^n = \text{sigmoid}(w_h^\top h_t^n + b_h), \tag{12}$$

$$q_t(n|\boldsymbol{x}, \boldsymbol{y}_{<t}) = \begin{cases} \chi_t^n \prod\limits_{n'<n} (1-\chi_t^{n'}), \text{ if } n<N \\ \prod\limits_{n'<N} (1-\chi_t^{n'}), \text{ otherwise} \end{cases} \tag{13}$$

where, $d$ is the dimension of the decoder states, $W_h \in \mathbb{R}^{N \times d}$ and $w_h \in \mathbb{R}^d$ are the weights of the halting mechanisms, and $b_h$ their biases. During inference the decoder exits when the halting signal $\chi_t^n$ exceeds a threshold $\tau_n$ which we tune on the valid set to achieve a better accuracy-speed trade-off. If thresholds $(\tau_n)_{1 \le n < N}$ have not been exceeded, then we default to exiting at block $N$.

The two classifiers are trained to minimize the cross-entropy with respect to either one the following oracle distributions:

**Likelihood-based:** At each time-step $t$, we choose the block whose exit classifier has the highest likelihood plus a regularization term weighted by $\lambda$ to encourage lower exits.

$$q_t^*(\boldsymbol{x}, \boldsymbol{y}) = \delta(\arg\max_n \text{LL}_t^n - \lambda n) \tag{14}$$

This oracle ignores the impact of the current decision on the future time-steps and we therefore consider smoothing the likelihoods with an RBF kernel.

$$\kappa(t, t') = e^{-\frac{|t-t'|^2}{\sigma}}, \quad \widetilde{\text{LL}_t^n} = \sum_{t'=1}^{|\boldsymbol{y}|} \kappa(t, t') \text{LL}_{t'}^n, \quad q_t^*(\boldsymbol{x}, \boldsymbol{y}) = \delta(\arg\max_n \widetilde{\text{LL}_t^n} - \lambda n), \tag{15}$$

where we control the size of the surrounding context with $\sigma$ the kernel width. We refer to this oracle as $\text{LL}(\sigma, \lambda)$ including the case where we only look at the likelihood of the current token with $\sigma \to 0$.

**Correctness-based:** Similar to the likelihood-based oracle we can look at the correctness of the prediction at time-step $t$ as well as surrounding positions. We define the target $q_t^*$ as follows:

$$C_t^n = \mathbb{1}[y_t = \arg\max_y p(y|h_{t-1}^n)], \quad \widetilde{C_t^n} = \sum_{t'=1}^{|\boldsymbol{y}|} \kappa(t, t') C_t^n, \tag{16}$$

$$q_t^*(\boldsymbol{x}, \boldsymbol{y}) = \delta(\arg\max_n \widetilde{C_t^n} - \lambda n). \tag{17}$$

**Confidence thresholding** Finally, we consider thresholding the model predictions (§2), i.e., exit when the maximum score of the current output classifier $p(y_{t+1}|h_t^n)$ exceeds a hyper-parameter threshold $\tau_n$. This does not require training and the thresholds $\boldsymbol{\tau} = (\tau_1, \ldots, \tau_{N-1})$ are simply tuned on the valid set to maximize BLEU. Concretely, for 10k iterations, we sample a sequence of thresholds $\boldsymbol{\tau} \sim \mathcal{U}(0,1)^{N-1}$, decode the valid set with the sampled thresholds and then evaluate the BLEU score and computational cost achieved with this choice of $\boldsymbol{\tau}$. After 10k evaluations we pick the best performing thresholds, that is $\boldsymbol{\tau}$ with the highest BLEU in each cost segment.

## 4 EXPERIMENTS

### 4.1 EXPERIMENTAL SETUP

We evaluate on several benchmarks and measure tokenized BLEU (Papineni et al., 2002):

**IWSLT'14 German to English (De-En).** We use the setup of Edunov et al. (2018) and train on 160K sentence pairs. We use $N = 6$ blocks, a feed-forward network (ffn) of intermediate-dimension

| | Uniform | $n = 1$ | $n = 2$ | $n = 3$ | $n = 4$ | $n = 5$ | $n = 6$ | Average |
|---|---|---|---|---|---|---|---|---|
| Baseline | - | 34.2 | 35.3 | 35.6 | 35.7 | 35.6 | 35.9 | 35.4 |
| Aligned ($\omega_n = 1$) | **35.5** | 34.1 | **35.5** | **35.8** | **36.1** | **36.1** | **36.2** | **35.6** |
| Mixed $M = 1$ | 34.1 | 32.9 | 34.3 | 34.5 | 34.5 | 34.6 | 34.5 | 34.2 |
| Mixed $M = 3$ | 35.1 | 33.9 | 35.2 | 35.4 | 35.5 | 35.5 | 35.5 | 35.2 |
| Mixed $M = 6$ | 35.3 | **34.2** | 35.4 | **35.8** | 35.9 | 35.8 | 35.9 | 35.5 |

Table 1: Aligned vs. mixed training on IWSLT De-En. We report valid BLEU for a uniformly sampled exit $n \sim \mathcal{U}([1..6])$ at each token, a fixed exit $n \in [1..6]$ for all tokens, as well as the average BLEU over the fixed exits. As baseline we show six standard Transformer models with 1-6 blocks.

1024, 4 heads, dropout 0.3, embedding dimension $d_{\text{enc}} = 512$ for the encoder and $d_{\text{dec}} = 256$ for the decoder. Embeddings are untied with 6 different output classifiers. We evaluate with a single checkpoint and a beam of width 5.

**WMT'14 English to French (En-Fr).** We also experiment on the much larger WMT'14 English-French task comprising 35.5m training sentence pairs. We develop on 26k held out pairs and test on newstest14. The vocabulary consists of 44k joint BPE types (Sennrich et al., 2016). We use a Transformer *big* architecture and tie the embeddings of the encoder, the decoder and the output classifiers ($(W_n)_{1 \leq n \leq 6}$; §2.1). We average the last ten checkpoints and use a beam of width 4.

Models are implemented in fairseq (Ott et al., 2019) and are trained with Adam (Kingma & Ba, 2015). We train for 50k updates on 128 GPUs with a batch size of 460k tokens for WMT'14 En-Fr and on 2 GPUs with 8k tokens per batch for IWSLT'14 De-En. To stabilize training, we re-normalize the gradients if the norm exceeds $g_{\text{clip}} = 3$.

For models with adaptive exits, we first train without exit prediction ($\alpha = 0$ in Eq. (7)) using the aligned mode (*c.f.* §2.2.1) for 50k updates and then continue training with $\alpha \neq 0$ until convergence. The exit prediction classifiers are parameterized by a single linear layer (Eq. (8)) with the same input dimension as the embedding dimension, *e.g.*, 1024 for a big Transformer; the output dimension is $N$ for a multinomial classifier or one for geometric-like. We exit when $\chi_{t,n} > 0.5$ for geometric-like classifiers.

## 4.2 TRAINING MULTIPLE OUTPUT CLASSIFIERS

We first compare the two training regimes for our model (§2.2). Aligned training performs self-attention on aligned states (§2.2.1) and mixed training exposes self-attention to hidden states from different blocks (§2.2.2).

We compare the two training modes when choosing either a uniformly sampled exit or a fixed exit $n = 1, \ldots, 6$ at inference time for every time-step. The sampled exit experiment tests the robustness to mixed hidden states and the fixed exit setup simulates an ideal setting where all previous states are available. As baselines we show six separate standard Transformers with $N \in [1..6]$ decoder blocks. All models are trained with an equal number of updates and mixed training with $M=6$ paths is most comparable to aligned training since the number of losses per sample is identical.

Table 1 shows that aligned training outperforms mixed training both for fixed exits as well as for randomly sampled exits. The latter is surprising since aligned training never exposes the self-attention mechanism to hidden states from other blocks. We suspect that this is due to the residual connections which *copy* features from lower blocks to subsequent layers and which are ubiquitous in Transformer models (§2). Aligned training also performs very competitively to the individual baseline models.

Aligned training is conceptually simple and fast. We can process a training example with $N$ exits in a single forward/backward pass while $M$ passes are needed for mixed training. In the remaining paper, we use the aligned mode to train our models. Appendix A reports experiments with weighing the various output classifiers differently but we found that a uniform weighting scheme worked well. On our largest setup, WMT'14 English-French, the training time of an aligned model with six output classifiers increases only marginally by about 1% compared to a baseline with a single output classifier keeping everything else equal.

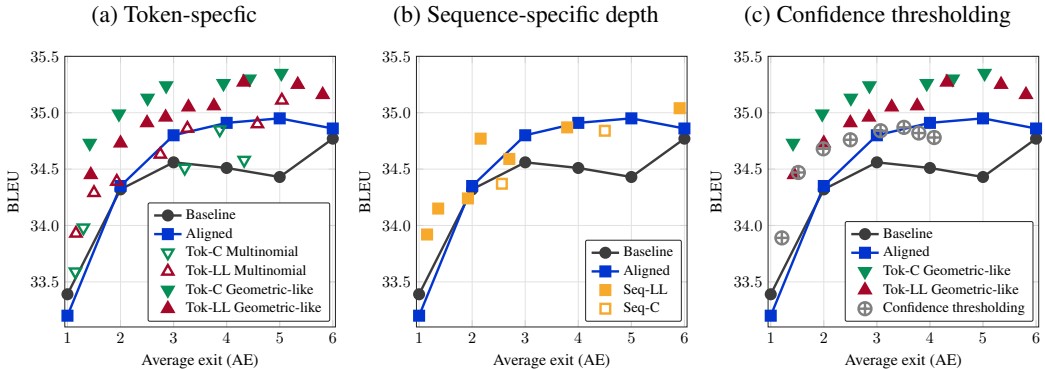

Figure 3: Trade-off between speed (average exit or AE) and accuracy (BLEU) for depth-adaptive methods on the IWSLT14 De-En test set.

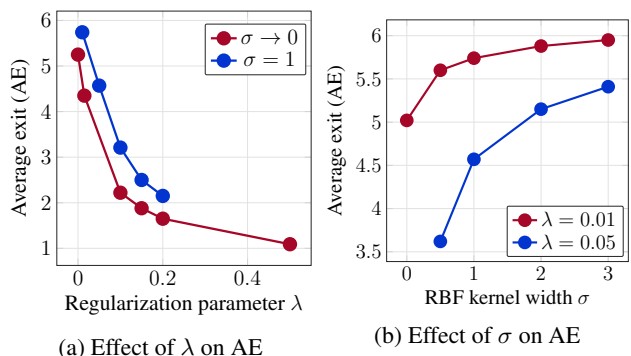

Figure 4: Effect of the hyper-parameters $\sigma$ and $\lambda$ on the average exit (AE) measured on the valid set of IWSLT'14 De-En.

### 4.3 ADAPTIVE DEPTH ESTIMATION

Next, we train models with aligned states and compare adaptive depth classifiers in terms of BLEU as well as computational effort. We measure the latter as the average exit per output token (AE).

As baselines we use again six separate standard Transformers with $N \in [1..6]$ with a single output classifier. We also measure the performance of the aligned mode trained model for fixed exits $n \in [1..6]$. For the adaptive depth token-specific models (Tok), we train four combinations: likelihood-based oracle (LL) + geometric-like, likelihood-based oracle (LL) + multinomial, correctness based oracle (C) + geometric-like and correctness-based oracle (C) + multinomial. Sequence-specific models (Seq) are trained with the correctness oracle (C) and the likelihood oracle (LL) with different values for the regularization weight $\lambda$. All parameters are tuned on the valid set and we report results on the test set for a range of average exits.

Figure 3 shows that the aligned model (blue line) can match the accuracy of a standard 6-block Transformer (black line) at half the number of layers ($n = 3$) by always exiting at the third block. The aligned model outperforms the baseline for $n = 2, \ldots, 6$.

For token specific halting mechanisms (Figure 3a) the geometric-like classifiers achieves a better speed-accuracy trade-off than the multinomial classifiers (filled vs. empty triangles). For geometric-like classifiers, the correctness oracle outperforms the likelihood oracle (Tok-C geometric-like vs. Tok-LL geometric-like) but the trend is less clear for multinomial classifiers. At the sequence-level, likelihood is the better oracle (Figure 3b).

The rightmost Tok-C geometric-like point ($\sigma = 0$, $\lambda = 0.1$) achieves 34.73 BLEU at AE = 1.42 which corresponds to similar accuracy as the $N = 6$ baseline at 76% fewer decoding blocks.

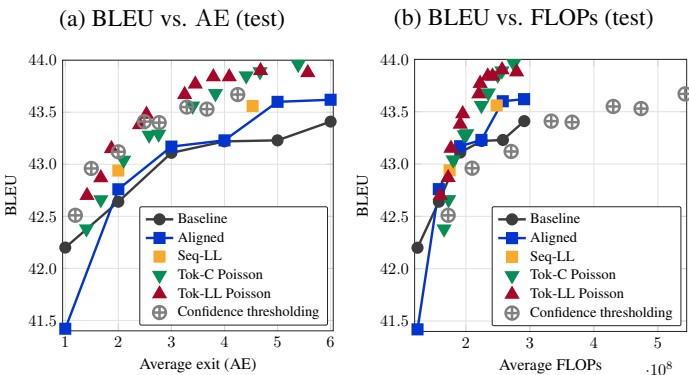

Figure 5: Speed and accuracy on the WMT'14 English-French benchmark (*c.f.* Figure 3).

The best accuracy of the aligned model is 34.95 BLEU at exit 5 and the best comparable Tok-C geometric-like configuration achieves 34.99 BLEU at AE = 1.97, or 61% fewer decoding blocks. When fixing the budget to two decoder blocks, Tok-C geometric-like with AE = 1.97 achieves BLEU 35, a 0.64 BLEU improvement over the baseline ($N = 2$) and aligned which both achieve BLEU 34.35.

Confidence thresholding (Figure 3c) performs very well but cannot outperform Tok-C geometric-like.

**Ablation of hyper-parameters** In this section, we look at the effect of the two main hyper-parameters on IWSLT'14 De-En: $\lambda$ the regularization scale (*c.f.* Eq. (9)), and the RBF kernel width $\sigma$ used to smooth the scores (*c.f.* Eq. (15)). We train Tok-LL Geometric-like models and evaluate them with their default thresholds (exit if $\chi_t^n > 0.5$). Figure 4a shows that higher values of $\lambda$ lead to lower exits. Figure 4b shows the effect of $\sigma$ for two values of $\lambda$. In both curves, we see that wider kernels favor higher exits.

## 4.4 SCALING THE ADAPTIVE-DEPTH MODELS

Finally, we take the best performing models form the IWSLT benchmark and test them on the large WMT'14 English-French benchmark. Results on the test set (Figure 5a) show that adaptive depth still shows improvements but that they are diminished in this very large-scale setup. Confidence thresholding works very well and sequence-specific depth approaches improve only marginally over the baseline. Tok-LL geometric-like can match the best baseline result of BLEU 43.4 ($N = 6$) by using only AE = 2.40 which corresponds to 40% of the decoder blocks; the best aligned result of BLEU 43.6 can be matched with AE = 3.25. In this setup, Tok-LL geometric-like slightly outperforms the Tok-C counterpart.

Confidence thresholding matches the accuracy of the $N=6$ baseline with AE 2.5 or 59% fewer decoding blocks. However, confidence thresholding requires computing the output classifier at each block to determine whether to halt or continue. This is a large overhead since output classifiers predict 44k types for this benchmark (§4.1). To better account for this, we measure the average number of FLOPs per output token (details in Appendix B). Figure 5b shows that the Tok-LL geometric-like approach provides a better trade-off when the overhead of the output classifiers is considered.

## 4.5 QUALITATIVE RESULTS

The exit distribution for a given sample can give insights into what a Depth-Adaptive Transformer decoder considers to be a difficult task. In this section, for each hypothesis $\widetilde{\boldsymbol{y}}$, we will look at the sequence of selected exits $(n_1, \ldots, n_{|\widetilde{\boldsymbol{y}}|})$ and the probability scores $(p_1, \ldots p_{|\widetilde{\boldsymbol{y}}|})$ with $p_t = p(\widetilde{y}_t | h_{t-1}^{n_t})$ *i.e.* the confidence of the model in the sampled token at the selected exit.

Figures 6 and 7 show hypotheses from the WMT'14 En-Fr and IWSLT'14 De-En test sets, respectively. For each hypothesis we state the exits and the probability scores. In Figure 6a, predicting

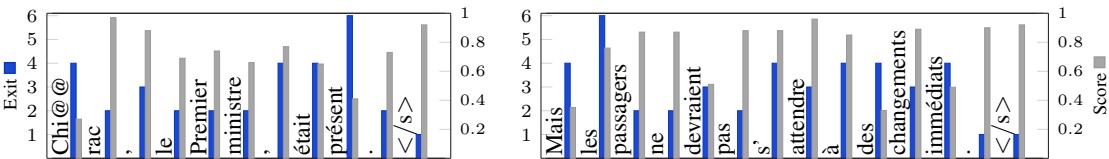

(a) **Src:** Chi@@rac , the Prime Minister , was there .
   **Ref:** Chi@@rac , Premier ministre , est là .

(b) **Src:** But passengers shoul@@dn't expect changes to happen immediately .
   **Ref:** Mais les passagers ne devraient pas s' attendre à des changements immédiats .

Figure 6: Examples from the WMT'14 En-Fr test set (newstest14) with Tok-LL geometric-like depth estimation. Token exits are in blue and confidence scores are in gray. The '@@' are due to BPE or subword tokenization. For each example the source (**Src**) and the reference (**Ref**) are provided in the caption.

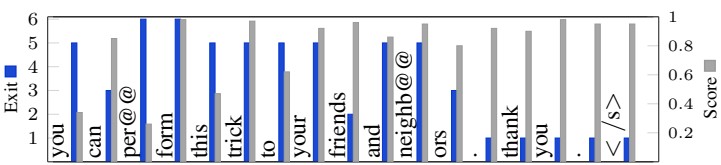

(a) **Src:** diesen trick können sie ihren freunden und nachbarn vor@@führen . danke .
   **Ref:** there is a trick you can do for your friends and neighb@@ors . thanks .

Figure 7: Example from the IWSLT'14 De-En test set with Tok-LL geometric-like depth estimation. See Figure 6 for more details.

'présent' (meaning 'present') is hard. A straightforward translation is 'était là' but the model chooses 'present' which is also appropriate. In Figure 6b, the model uses more computation to predict the definite article 'les' since the source has omitted the article for 'passengers'.

A clear trend in both benchmarks is that the model requires less computation near the end of decoding to generate the end of sequence marker $$ and the preceding full-stop when relevant. In Figure 8, we show the distribution of the exits at the beginning and near the end of test set hypotheses. We consider the beginning of a sequence to be the first 10% of tokens and the end as the last 10% of tokens. The exit distributions are shown for three models on WMT'14 En-Fr: $Model_1$ has an average exit of $AE = 2.53$, $Model_2$ exits at $AE = 3.79$ on average and $Model_3$ with $AE = 4.68$. Within the same models, deep exits late are used at the beginning of the sequence and early exits are selected near the end. For heavily regularized models such as $Model_1$ with $AE = 2.53$, the disparity between beginning and end is less severe as the model exits early most of the time. $Model_2$ and $Model_3$ are less regularized (higher AE) and tend to use late exits at the beginning of the sequence and early exits near the end. On the other hand, the more regularized $Model_1$ with $AE = 2.53$ exits

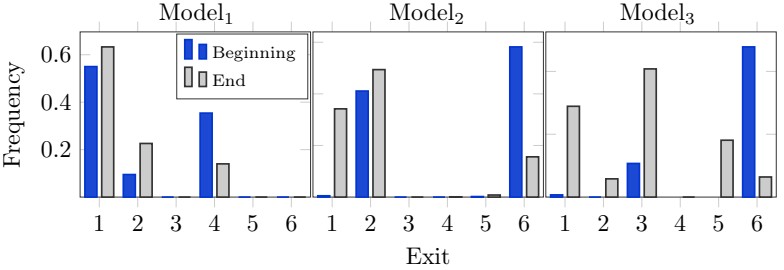

Figure 8: WMT'14 En-Fr test set: exit distributions in the beginning (relative-position: rpos<0.1) and near the end (rpos>0.9) of the hypotheses of three models.

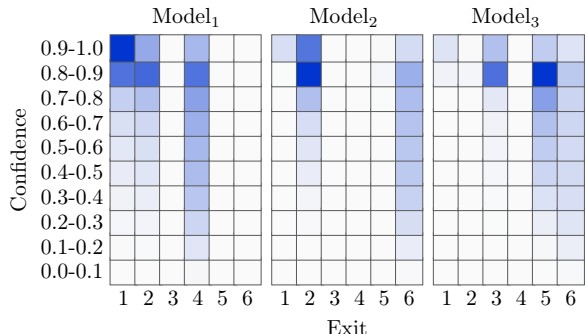

Figure 9: Joint histogram of the exits and the confidence scores for 3 Tok-LL geometric-like models on newstest14.

early most of the time. There is also a correlation between the model probability and the amount of computation, particularly in models with low $\mathrm{AE}$. Figure 9 shows the joint histogram of the scores and the selected exit. For both $\mathrm{Model}_1$ and $\mathrm{Model}_2$, low exits ($n \leq 2$) are used in the high confidence range $[0.8 - 1]$ and high exits ($n \geq 4$) are used in the low-confidence range $[0 - 0.5]$. $\mathrm{Model}_3$ has a high average exit ($\mathrm{AE} = 4.68$) so most tokens exit late, however, in low confidence ranges the model does not exit earlier than $n = 5$.

## 5 CONCLUSION

We extended anytime prediction to the structured prediction setting and introduced simple but effective methods to equip sequence models to make predictions at different points in the network. We compared a number of different mechanisms to predict the required network depth and find that a simple correctness based geometric-like classifier obtains the best trade-off between speed and accuracy. Results show that the number of decoder layers can be reduced by more than three quarters at no loss in accuracy compared to a well tuned Transformer baseline.

### ACKNOWLEDGMENTS

We thank Laurens van der Maaten for fruitful comments and suggestions.

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

## APPENDIX A    LOSS SCALING

In this section we experiment with different weights for scaling the output classifier losses. Instead of uniform weighting, we bias towards specific output classifiers by assigning higher weights to their losses. Table 2 shows that weighing the classifiers equally provides good results.

|  | Uniform | $n=1$ | $n=2$ | $n=3$ | $n=4$ | $n=5$ | $n=6$ | Average |
|---|---|---|---|---|---|---|---|---|
| Baseline | - | 34.2 | 35.3 | 35.6 | 35.7 | 35.6 | 35.9 | 35.4 |
| $\omega_n=1$ | 35.5 | 34.1 | **35.5** | **35.8** | **36.1** | 36.1 | 36.2 | **35.6** |
| $\omega_n=n$ | 35.3 | 32.2 | 35.0 | **35.8** | 36.0 | **36.2** | **36.3** | 35.2 |
| $\omega_n=\sqrt{n}$ | 35.4 | 33.3 | 35.2 | **35.8** | 35.9 | 36.1 | 36.1 | 35.4 |
| $\omega_n=1/\sqrt{n}$ | **35.6** | 34.5 | 35.4 | 35.7 | 35.8 | 35.8 | 35.9 | 35.5 |
| $\omega_n=1/n$ | 35.3 | **34.7** | 35.3 | 35.5 | 35.7 | 35.8 | 35.8 | 35.5 |

(a) IWSLT De-En - Valid

|  | Uniform | $n=1$ | $n=2$ | $n=3$ | $n=4$ | $n=5$ | $n=6$ | Average |
|---|---|---|---|---|---|---|---|---|
| Baseline | - | 33.7 | 34.6 | 34.6 | 34.6 | 34.6 | 34.8 | 34.5 |
| $\omega_n=1$ | 34.4 | 33.2 | **34.4** | **34.8** | **34.9** | **35.0** | 34.9 | **34.5** |
| $\omega_n=n$ | 34.2 | 31.4 | 33.8 | 34.7 | 34.8 | 34.8 | 34.9 | 34.1 |
| $\omega_n=\sqrt{n}$ | 34.4 | 32.5 | 34.1 | **34.8** | **34.9** | **35.0** | **35.1** | 34.4 |
| $\omega_n=1/\sqrt{n}$ | 34.6 | 33.7 | 34.3 | 34.6 | 34.8 | 34.8 | 34.9 | **34.5** |
| $\omega_n=1/n$ | 34.2 | **33.8** | 34.3 | 34.5 | 34.6 | 34.7 | 34.7 | 34.4 |

(b) IWSLT De-En - Test

Table 2: Aligned training with different weights ($\omega_n$) on IWSLT De-En. For each model we report BLEU on the dev set evaluated with a uniformly sampled exit $n \sim \mathcal{U}([1..6])$ for each token and a fixed exit $n \in [1..6]$ throughout the sequence. The average corresponds to the average BLEU over the fixed exits.

**Gradient scaling**    Adding intermediate supervision at different levels of the decoder results in richer gradients for lower blocks compared to upper blocks. This is because earlier layers affect more loss terms in the compound loss of Eq. (4). To balance the gradients of each block in the decoder, we scale up the gradients of each loss term $(-\text{LL}_n)$ when it is updating the parameters of its associated block ($\text{block}_n$ with parameters $\theta_n$) and revert it back to its normal scale before back-propagating it to the previous blocks. Figure 10 and Algorithm 1 illustrate this gradient scaling procedure. The $\theta_n$ are updated with $\gamma_n$-amplified gradients from the block's supervision and $(N-n)$ gradients from the subsequent blocks. We choose $\gamma_n = \gamma(N-n)$ to control the ratio $\gamma{:}1$ as the ratio of the block supervision to the subsequent blocks' supervisions.

Table 3 shows that gradient scaling can benefit the lowest layer at the expense of higher layers. However, no scaling generally works very well.

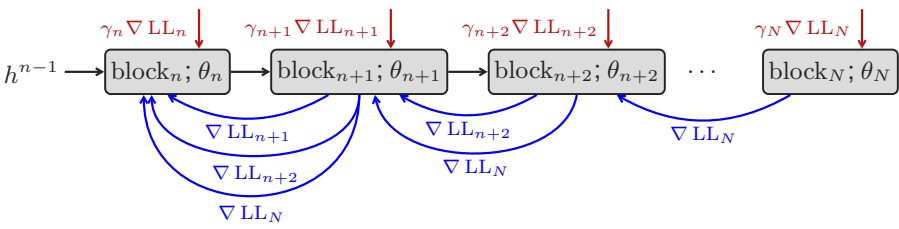

Figure 10: Illustration of gradient scaling.

---

**Algorithm 1** Pseudo-code for gradient scaling (illustrated for a single step $t$)

---

1: **for** $n \in 1..N$ **do**
2: $\quad h_t^n = \text{block}_n(h_t^{n-1})$
3: $\quad p(y_{t+1}|h_t^n) = \text{softmax}(W_n h_t^n)$
4: $\quad p(y_{t+1}|h_t^n) = \text{SCALE\_GRADIENT}(p(y_{t+1}|h_t^n), \gamma_n)$
5: $\quad$ **if** $n < N$ **then** $h_t^n = \text{SCALE\_GRADIENT}(h_t^n, \dfrac{1}{\gamma_{n+1}})$
6: **end for**
7: **function** SCALE\_GRADIENT(Tensor $x$, scale $\gamma$)
8: $\quad$ **return** $\gamma x + (1 - \gamma)\text{STOP\_GRADIENT}(x)$
9: $\quad \triangleright$ STOP\_GRADIENT in PyTorch with $x$.detach().
10: **end function**

---

|  | Uniform | $n=1$ | $n=2$ | $n=3$ | $n=4$ | $n=5$ | $n=6$ | Average |
|---|---|---|---|---|---|---|---|---|
| Baseline | - | 34.2 | 35.3 | 35.6 | 35.7 | 35.6 | 35.9 | 35.4 |
| $\emptyset$ | **35.5** | 34.1 | **35.5** | **35.8** | **36.1** | **36.1** | **36.2** | **35.6** |
| $\gamma = 0.3$ | 35.1 | 33.7 | 34.7 | 35.3 | 35.7 | 35.8 | 36.0 | 35.2 |
| $\gamma = 0.5$ | 35.4 | 34.8 | 35.4 | 35.6 | 35.6 | 35.7 | 35.6 | 35.4 |
| $\gamma = 0.7$ | 34.9 | 34.6 | 35.1 | 35.1 | 35.2 | 35.4 | 35.3 | 35.1 |
| $\gamma = 0.9$ | 34.9 | 34.8 | 35.3 | 35.3 | 35.3 | 35.4 | 35.5 | 35.3 |
| $\gamma = 1.1$ | 35.1 | **34.9** | 35.2 | 35.3 | 35.3 | 35.3 | 35.3 | 35.2 |

(a) IWSLT De-En - Valid

|  | Uniform | $n=1$ | $n=2$ | $n=3$ | $n=4$ | $n=5$ | $n=6$ | Average |
|---|---|---|---|---|---|---|---|---|
| Baseline | - | 33.7 | 34.6 | 34.6 | 34.6 | 34.6 | 34.8 | 34.5 |
| $\emptyset$ | 34.4 | 33.2 | **34.4** | **34.8** | **34.9** | **35.0** | 34.9 | 34.5 |
| $\gamma = 0.3$ | 34.2 | 32.8 | 33.9 | 34.3 | 34.6 | 34.8 | **35.0** | 34.2 |
| $\gamma = 0.5$ | **34.5** | 33.8 | 34.2 | 34.6 | 34.5 | 34.7 | 34.7 | **34.6** |
| $\gamma = 0.7$ | 34.0 | 33.7 | 34.2 | 34.3 | 34.3 | 34.3 | 34.3 | 34.2 |
| $\gamma = 0.9$ | 34.1 | **34.0** | 34.2 | 34.3 | 34.4 | 34.4 | 34.4 | 34.3 |
| $\gamma = 1.1$ | 34.2 | **34.0** | 34.3 | 34.3 | 34.3 | 34.3 | 34.2 | 34.2 |

(b) IWSLT De-En - Test

Table 3: Aligned training with different gradient scaling ratios $\gamma : 1$ on IWSLT'14 De-En. For each model we report the BLEU4 score evaluated with a uniformly sampled exit $n \sim \mathcal{U}([1..6])$ for each token and a fixed exit $n \in [1..6]$. The average corresponds to the average BLEU4 of all fixed exits.

## APPENDIX B  FLOPS APPROXIMATION

This section details the computation of the FLOPS we report. The per token FLOPS are for the decoder network only since we use an encoder of the same size for all models. We breakdown the FLOPS of every operation in Algorithm 2 (blue front of the algorithmic statement). We omit non-linearities, normalizations and residual connections. The main operations we account for are dot-products and by extension matrix-vector products since those represent the vast majority of FLOPS (we assume batch size one to simplify the calculation).

| Parameters | |
| --- | --- |
| $d_d$ | decoder embedding dimension. |
| $d_e$ | encoder embedding dimension. |
| $d_f$ | The feed-forward network dimension. |
| $|\boldsymbol{x}|$ | source length. |
| $t$ | Current time-estep ($t \geq 1$). |
| $V$ | output vocabulary size. |

| Operation | FLOPS |
| --- | --- |
| Dot-product ($d$) | $2d - 1$ |
| Linear $d_{in} \to d_{out}$ | $2d_{in}d_{out}$ |

Table 4: FLOPS of basic operations, key parameters and variables for the FLOPS estimation.

With this breakdown, the total computational cost at time-step $t$ of a decoder block that we actually go through, denoted with FC, is:

$$\mathrm{FC}(\boldsymbol{x}, t) = 12d_d^2 + 4d_f d_d + 4t d_d + 4|\boldsymbol{x}|d_d + 4[\![\mathrm{FirstCall}]\!]|\boldsymbol{x}|d_d d_e,$$

where the cost of mapping the source' keys and values is incurred the first time the block is called (flagged with FirstCall). This occurs at $t = 1$ for the baseline model but it is input-dependent with depth adaptive estimation and may never occur if all tokens exit early.

If skipped, a block still has to compute the keys and value of the self-attention block so the self-attention of future time-steps can function. We will denote this cost with FS and we have $\mathrm{FS} = 4d_d^2$.

Depending on the halting mechanism, an exit prediction cost, denoted wit FP, is added:

$$
\begin{aligned}
\text{Sequence-specific depth:} \quad & \mathrm{FP}(t, q(t)) = 2[\![t = 1]\!]N d_d \\
\text{Token-specific Multinomial:} \quad & \mathrm{FP}(t, q(t)) = 2N d_d \\
\text{Token-specific Geometric-like:} \quad & \mathrm{FP}(t, q(t)) = 2d_d q(t) \\
\text{Confidence thresholding:} \quad & \mathrm{FP}(t, q(t)) = 2q(t)V d_d
\end{aligned}
$$

For a set of source sequences $\{\boldsymbol{x}^{(i)}\}_{i \in \mathcal{I}}$ and generated hypotheses $\{\boldsymbol{y}^{(i)}\}_{i \in \mathcal{I}}$, the average flops per token is:

$$
\begin{aligned}
\text{Baseline ($N$ blocks):} \quad & \frac{1}{\sum_i |\boldsymbol{y}^{(i)}|} \sum_i \sum_{t=1}^{|\boldsymbol{y}^{(i)}|} \left( N \, \mathrm{FC}(\boldsymbol{x}^{(i)}, t) + 2V d_d \right) \\
\text{Adaptive depth:} \quad & \frac{1}{\sum_i |\boldsymbol{y}^{(i)}|} \sum_i \sum_{t=1}^{|\boldsymbol{y}^{(i)}|} \left( q(t)\mathrm{FC}(\boldsymbol{x}^{(i)}, t) + (N - q(t))\mathrm{FS} + \mathrm{FP}(t, q(t)) + 2V d_d \right)
\end{aligned}
$$

In the case of confidence thresholding the final output prediction cost ($2V d_d$) is already accounted for in the exit prediction cost FP.

---

**Algorithm 2** Adaptive decoding with Tok-geometric-like

---

1: **Input:** source codes $s$, incremental state
2: **Initialization:** $t = 1$, $y_1 = $
3: **for** $n \in 1 \ldots N$ **do**
4:     FirstCall[$n$] = True.    ▷ A flag signaling if the source' keys and values should be evaluated.
5: **end for**
6: **while** $y_t \neq $ **do**
7:     Embed the last output token $y_t$.
8:     **for** $n \in 1 \ldots N$ **do**
9:         ▷ **Self-attention.**
10:         - Map the input into a key ($k$) and value ($v$). FLOPS=$4d_d^2$
11:         - Map the input into a query $q$. FLOPS=$2d_d^2$
12:         - Score the memory keys with $q$ to get the attention weights $\alpha$. FLOPS=$4td_d$
13:         - Map the attention output. FLOPS=$2d_d^2$
14:         ▷ **Encoder-Decoder interaction.**
15:         **if** FirstCall[$n$] **then**
16:             Map the source states into keys and values for the nth block. FLOPS=$4|\boldsymbol{x}|d_e d_d$
17:             FirstCall[$n$] = False
18:         **end if**
19:         - Map the input into a query $q$. FLOPS=$2d_d^2$
20:         - Score the memory keys with $q$ to get the attention weights $\alpha$. FLOPS=$4|\boldsymbol{x}|d_d$
21:         - Map the attention output. FLOPS=$2d_d^2$
22:         Feed-forward network. FLOPS=$4d_d d_f$
23:         Estimate the halting probability $\chi_{t,n}$. FLOPS=$2d_d$
24:         **if** $\chi_{t,n} > 0.5$ **then**
25:             Exit the loop (Line 8)
26:         **end if**
27:     **end for**
28:     **if** $n < N$ **then**
29:         ▷ **Skipped blocks.**
30:         **for** $n_s \in n + 1 \ldots N$ **do**
31:             Copy and map the copied state into a key ($k$) and value ($v$). FLOPS=$4d_d^2$
32:         **end for**
33:     **end if**
34:     Project the final state and sample a new output token. FLOPS=$2Vd_d$
35:     $t++$
36: **end while**

---

