# OpenReview forum: "Depth-Adaptive Transformer"
_ICLR.cc/2020/Conference — Accept (Poster)_

### Official Review · AnonReviewer3 · 2019-10-07
**Official Blind Review #3**

**Rating:** 3

**Review:**

[Post-reponse EDIT]: After reading the authors' response, and taking into account the other reviewers' opinions, I am willing to suggest acceptance of this paper on the condition that ***the authors will add the experimental results to address issue #7 and #8 of my original review (as they promised in their rebuttal)***. Before that, I'm keeping the current score, while noting that a more accurate rating should be "5 - Marginally below the acceptance threshold" (which is not available in ICLR this year). I want to thank the authors for addressing the other issues/questions I raised.

----------------------------------------- Original Review Below -----------------------------------------

Summary: The author proposes a new way to improve the supervision of training encoder-decoder Transformers so that it can make flexible, depth-adaptive predictions at inference time. The paper compares multiple dynamic computation schemes to investigate the effectiveness of these different approaches (e.g., at both sequence and token levels). Relatively extensive experiments were conducted in different settings (on a small and a large MT dataset).

While the dynamic halting on Transformers has been previously explored in the Universal Transformers paper, this technique has not been applied to standard (encoder-)decoder Transformers. I think this is an interesting and exciting topic to pursue, as the authors seem to demonstrate (at least on the small IWSLT'14 DE-EN dataset) that we may not need as many layers at the decoding phase as we thought (and thus # of FLOPs).

However, I think the paper can be further improved in terms of both its organization/clarity and its experimental study (see details following).

--------------------------------------------

Questions/comments:

1. Please add a Related Work section. While using dynamic halting on standard Transformers may be novel, general dynamic routing on deep networks (which also aims to reduce the computation) is not. For example, SkipNet [1] uses an input-based dynamic skipping to bypass certain layers of a DNN. It's a good scholarship to give proper credit to prior related work(s).

2. I think Poisson binomial (PB) may be a misnomer for the second qt modeling technique. PB is a probability distribution on the __sum__ of N (unequal) Bernoulli trials [2], which doesn't make sense here. What you are looking for, I think, is a learned "coin flip" at every layer of the network to decide whether to continue to the next layer (i.e., you are looking for the first "head" in the coin flips, instead of the sum of heads). And indeed, in Eq. (9), the definition for qt(n|...) is a "geometric-alike" distribution with success probability \chi_{t,n} at the n^th flip. (Which brings to a side point: for the "otherwise" case in Eq. (9), I think you mean \prod_n (1-\chi_{t, n}); otherwise you can't guarantee \sum_n qt(n|...) = 1 :-) )

3. Why and how did you pick the exit threshold \chi_{t, n} > 0.5 for your experiments? According to your definition of \chi_{t,n} and q_t in Eq. (9), isn't the correct thing to do to sample from \chi_{t, n}? My major concern is that, given different \sigma (which you didn't specify, but I guess is something like a sigmoid), your \chi_{t, n} may land in very different ranges. What if you have the following case: \chi_{t, 1} = 0.501, \chi_{t, 2} = 0.1, \chi_{t,3} = 0.999, \chi_{t,4} = 0.95? Doesn't this mean it may worth it to stop at the 3rd layer instead? (Using the coin flip analogy, having a coin with head probability 0.501 doesn't necessarily mean you "must" get a "head" at the first flip.) Moreover, if I use $\sigma = sigmoid(x)/2 + 0.5$, won't I always get a \chi > 0.5?

4. In the token-specific likelihood-based method, you used an RBF kernel to model the influence of a time step t on its neighboring time steps. Two questions: 1) When you do \sum_{t'}, did you also include t'<t? 2) Have you tried any other kernels, and how does this choice affect the performance?

5. The aligned training is actually a type of intermediate auxiliary loss (or deep supervision, as the computer vision community probably more often calls it), which makes it not surprising that the unaligned Transformer could perform slightly better than the baseline. I find it interesting that the mixed training strategy doesn't work.

6. Why did you use different training schemes for WMT'14 and IWSLT'14 (e.g., freezing the parameters, etc.)?

7. One of the major concerns I have is that the empirical results don't seem that impressive. First, to demonstrate the effectiveness of the proposed adaptive depth estimation methodology, besides comparing to the baseline (black line) in Figures 3 & 4, you should also compare with the blue line (which adds these auxiliary losses to the baseline, but doesn't use the adaptive strategy that involves qt(n|...) at all) for fairness. From Figures 3 & 4, it seems that the adaptive-depth predictions are usually in the close neighborhood of the blue lines when at the same value of AE (instead of substantial, consistent improvement). In addition, while Tok-LL Poisson did well in Figure 4(b), it didn't seem to make a difference in Figure 4(a), which is where one is supposed to tune the hyperparameters on (such as \lambda, \sigma; see #8 below). This makes me a bit dubious about how much help the adaptive module brought, compared to just using the aligned model.

8. For the WMT'14 experiments (Figure 4), why is there only one run of the "Tok-LL Poisson"? My understanding is that, if you are to use the proposed approach on a new dataset, you would want to try different settings of (\sigma, \lambda) on the validation set and pick the best one to use for testing. In Figure 4(a), for instance, for Seq-LL I would probably pick the (\sigma, \lambda) setting corresponding to the top-right yellow square--- which turns out to be slightly worse than the blue line on the test set (Figure 4(b)). It'd be useful to plot more "red triangles" in Figure 4 to evaluate how much the adaptive-depth methodology contributes to the performance.

========================

Some minor errors that don't have much impact on the score:

9. The beginning sentence of the 4th paragraph of Section 1 is a bit strange (grammatically).

10. Not all equations are numbered! See section 2.

11. In the first equation of Section 2, do you mean h_{\leq t}^{n-1} rather than h_{< t}^{n-1}? I think h_t from the previous layer is also used?

12. In the 3rd equation of Eq. (12), you should have m somewhere within the summation.

13. Equation (7) missing a right parenthesis.

14. In Eq. (8) W_h is a matrix, whereas in Eq. (9) W_h is a vector (if I'm not mistaken)? Maybe use a lowercase w.

15. Inconsistent notations. For instance, as I described in #3 above, you didn't say in the paper what \sigma means in Eq. (9), but later re-used the same letter for a different meaning in Eq. (11) for the RBF kernel. Another example is the usage of \theta_n in the "Confidence thresholding" paragraph for the threshold value; you used the same letter again in the "Gradient scaling" section of the appendix but with a different meaning (learnable parameters).

16. In the last sentence of the Likelihood-based token specific depth, I'd suggest \sigma \rightarrow 0 instead of \sigma=0, which would otherwise make \frac{...}{\sigma} in Eq. (11) undefined.

17. I don't think you specified the \theta_n used in your experiments (and how you tune them).

18. What is the FS in Appendix B? Also, for "ffn" in Table 4, it's best to write its full name before referring to it with acronyms.

19. Just curious: how did you implement the gradient scaling described in Appendix A (e.g., ensuring the \gamma_n only applies to block n, but not blocks < n)?

========================

I think the current shape of the paper is marginally below the acceptance threshold. (Note that while the rating is "3 - Weak reject", I don't mean the score, but only the "weak reject" part. I'm still excited about the idea of applying dynamic computation in Transformer.) But I'm happy to consider adjusting the score if my concerns above can be satisfactorily addressed.


[1] https://arxiv.org/abs/1711.09485
[2] https://en.wikipedia.org/wiki/Poisson_binomial_distribution

**Experience Assessment:**

I have published one or two papers in this area.

**Review Assessment: Checking Correctness Of Derivations And Theory:**

N/A

**Review Assessment: Checking Correctness Of Experiments:**

I carefully checked the experiments.

**Review Assessment: Thoroughness In Paper Reading:**

I read the paper at least twice and used my best judgement in assessing the paper.

---

> ### Author Response · Authors · 2019-11-14
> **Thank you for your detailed and insightful comments!**
>
>
> Thank you for your detailed and insightful comments! Please see our response below.
>
> >> 1. Skipnet & other related work
> We cited this related work in the updated version of the paper just posted.
>
>
> >> 2. I think Poisson binomial may be a misnomer
>
> This is a good point! We corrected eq (9), it is indeed $\prod_{n’=1}^N  (1-{\chi_t}^{n’})$ for n=N.
> It is similar to a geometric distribution but with different success rates chi_t^n.
> We changed the name of the distribution to avoid any misunderstanding.
>
>
> >> 3. Why and how did you pick the exit threshold $\chi_{t, n} > 0.5$?
>
> Sigma is the sigmoid function. We choose threshold 0.5 to simply select the likely outcome  (continue or stop), i.e., we choose to follow the argmax of the distribution. We did experiment with tuning the threshold(s) similar to confidence thresholding which yielded slightly better results but the 0.5 threshold was a good compromise between simplicity and performance. Sampling a decision is an interesting avenue for future work. We focused on different ways to modeling the distribution itself.
>
>
> >> 4. RBF kernel: When you do $\sum_{t'}$, did you also include t'<t? Have you tried any other kernels, and how does this choice affect the performance?
>
> Yes, we also include t’<t in the summation.
> We did not try other kernels. The motivation for the RBF kernel is to smooth the scores of each step by including signals from neighbouring tokens. As for the effect of sigma on the performance, we show an ablation study of these parameters in Appendix C of the updated paper.
>
>
> >> 5. The aligned training is a type of intermediate auxiliary loss (or deep supervision, as the CV community probably more often calls it), which makes it not surprising that the unaligned Transformer could perform slightly better than the baseline. I find it interesting that the mixed training strategy doesn't work.
>
> Yes, aligned training is similar to deep supervision in CV. However, it is somewhat surprising that it also worked when we used different exits within the same sequence at inference. This exposes the self-attention to copied states - something that did not happen during training.
>
> Mixed training does work, it performs as well as training N different models of varying depth, cf. Table 1 “baseline” vs. “Mixed M=6” (most comparable setup to baseline). However, the aligned training does perform slightly better than both. The mixed training is also more costly to training since M different forward/backwards are required per sample.
>
>
> >> 6. Why did you use different training schemes for WMT'14 and IWSLT'14 (e.g., freezing the parameters, etc.)?
>
> We follow previous work on the number of checkpoints we use to average (WMT En-Fr) or no averaging at all (IWSLT De-En). WMT En-Fr requires ten checkpoints, epochs are long since it is a large dataset, and model performance did deteriorate when the exit classifier was trained for too long. Therefore, we opted to freeze the encoder-decoder classifier parameters for this dataset only.
>
>
> >> 7. One of the major concerns I have is that the empirical results don't seem that impressive.
> >> ... you should also compare with the blue line
>
> The aligned model (blue line) is part of our approach, hence the comparison to the baseline. Aligned is a single model while the baseline consists of N=6 different models which is a clear advantage of the aligned approach that is part of our method.
>
> >> ... it seems that the adaptive-depth predictions are usually in the close neighborhood of the blue lines when at the same value of AE ...
>
> This is true for higher AE but there are substantial improvements for low average exits (AE): Fig 3(a) & (b) show improvements of 0.5 BLEU for AE 1-1.5. Therefore, when we really would like to save a lot of computation, adaptive depth can make a meaningful impact.
>
>
> >> In addition, while Tok-LL Poisson did well in Figure 4(b), it didn't seem to make a difference in Figure 4(a), which is where one is supposed to tune the hyper-parameters on (such as $\lambda$, $\sigma$; see #8 below).
>
> The results in Fig 4 are not well tuned because we simply ran out of time until the deadline; see the response for #8 below.
>
>
> >> 8. For the WMT'14 experiments (Fig 4), why is there only one run of the "Tok-LL Poisson"?
>
> To meet the submission deadline, we only had time to train a single model and optimize a single $(\sigma, \lambda)$ configuration for WMT En-Fr.
>
>
> Minor comments:
>
> Thank you for the thorough feedback, we fixed the mistakes you pointed out in the updated version.
>
> >> 17. I don't think you specified the $\theta_n$ used in your experiments (and how you tune them).
> We added the details for confidence thresholding in Appendix B of the updated version.
>
> >> 19. Just curious: how did you implement the gradient scaling described in Appendix A
> We added the details for this in Appendix A of the updated version.

---

> > ### Comment · AnonReviewer3 · 2019-11-15
> > **Further comments on the author's response**
> >
> > Thank you for the responses. I'm glad to see that some of my questions are resolved (e.g., the PB issue and the kernel issue). And I appreciate the efforts the authors put in updating the paper to fix the various typos and ambiguities.
> >
> > I am still disappointed that the authors are unable to fully address my concerns in the part of the experiment.
> >
> > 1) Regarding #7, the blue line is indeed "partly your approach". But as I said, the blue line is also simply doing deep supervision, whose comparison to the baseline does not come at a surprise to me (even if you may take different exits, as you indicated, you still have __already explicitly optimized__ for it using the auxiliary loss at training time). Moreover, I feel the claim that "Fig 3(a) & (b) show improvements of 0.5 BLEU for AE 1-1.5" is still not strong enough to prove its usefulness, because i) for "aligned" networks, you can only pick, discretely, either layer # = 1 or 2. There is no middle ground (e.g., doesn't make sense to say what "1.5 layer" means in this setting). The blue line segment connecting AE=1 and AE=2 is largely a part of the line chart (for the purpose to connect dots). ii) 2 layers are not deep. Once you want to balance performance with a "reasonably" early exit, I think AE=2 is not too bad at all. However, the red triangles are indeed around the blue line when AE=2, in all of the figures.
> >
> > 2) Regarding the "not enough time" to further study Figure 4. Is the reported "Tok-LL Poisson" setting (and the $\lambda, \sigma$ setting) the only one you have ever tried while investigating the approach? (I thought you also froze the encoder-decoder parameters from the first phase to make training more efficient, wouldn't that enable you to explore more settings, too?) Did you manage to start running anything since the response period begins which may be reportable at this stage? If so, it'd be super helpful if you can update your draft with the latest results. I am still not convinced with the current results, since WMT'14 is a major large-scale dataset.
> >
> > As I highlighted in the original review, #7 and #8 are the major concerns I had (and they still are). I'd appreciate it if the authors can provide more empirical evidence and study of different settings in their paper. I look forward to reading the (final) revised draft soon.

---

> > > ### Author Response · Authors · 2019-11-15
> > > **Re: Further comments on the author's response**
> > >
> > > >> the blue line is also simply doing deep supervision, whose comparison to the baseline does not come at a surprise to me (even if you may take different exits, as you indicated, you still have __already explicitly optimized__ for it using the auxiliary loss at training time).
> > >
> > > We respectfully disagree with the point that the aligned training perfectly optimizes for different exits. In sequence modeling the self-attention requires all hidden states of the previous time steps for the previous layer. Aligned training assumes this is always available but it is simply not the case at inference time. This is a clear train/test mismatch. The states the self-attention is exposed to at training and inference time differ and this is not addressed by training multiple exits.
> > >
> > >
> > > >>  Moreover, I feel the claim that "Fig 3(a) & (b) show improvements of 0.5 BLEU for AE 1-1.5" is still not strong enough to prove its usefulness, because i) for "aligned" networks, you can only pick, discretely, either layer # = 1 or 2. There is no middle ground (e.g., doesn't make sense to say what "1.5 layer" means in this setting).
> > >
> > > Yes, the comparison between integer values of AE is not straightforward as you say, however, this is an artifact of the baselines. Fig 3(a) shows a clear improvement of 0.5 BLEU for AE=1.1 - a point that is very close to AE=1 of the blue line. There are also improvements in Fig. 3a/b for AE 4-5.
> > >
> > >
> > > >> I'd appreciate it if the authors can provide more empirical evidence and study of different settings in their paper.
> > >
> > > We are working on these experiments and hope to be able to update the paper soon but the results won't be ready until the rebuttal period is over.

---

> > > > ### Comment · AnonReviewer3 · 2019-11-15
> > > > **Further "further comments" on the authors' response**
> > > >
> > > > Thanks for your response. I think empirical studies in more settings on the large-scale benchmark would be important for future versions of this paper. So I'm glad that you have started to run it.
> > > >
> > > > Regarding #1 above: I never said anything about "perfectly optimizes" :-) But I did say that adding auxiliary loss makes it a more "explicitly" supervised process (of what exit at layer 2/3/4/5 should do). If you imagine a 100-layer network with 99 auxiliary losses injected to its intermediate layers, then intuitively these auxiliary losses are (likely) encouraging the outputs of each layer to be similar (to the 100^th layer). Lower layers may not be as good as deeper layers; but they are **supervised explicitly** to be similar.  Therefore, while the training and testing settings are different indeed, I still don't think it's a surprise to me.
> > > >
> > > > Regarding #2 above: "There are also improvements in Fig. 3a/b for AE 4-5". Indeed. But there are also points where it seems to be not as good for AE 3-5. And that is exactly the problem. It doesn't seem like convincing enough evidence to reject the "null hypothesis" here: that your method is not superior to the blue line.

---

### Official Review · AnonReviewer2 · 2019-10-23
**Official Blind Review #2**

**Rating:** 6

**Review:**

The authors study depth-adaptive Transformer models and show that they can perform well even under fairly basic strategies to stop the Transformer early. The paper is well written and the results convincing. There is not a lot of novelty but the results hold well, so it is a clear contribution. One main issue that prevents this reviewer from increasing the rating is the measure of speed the authors use: counting exit at every token. This is a fine measure for inference time, but during training the model proceeds on the whole sequence, right? Could you provide speed numbers for the training step? Is there any improvement over a baseline Transformer or is this technique solely for inference? (Which is still a contribution, but it should be made clear in the paper.)

I thank the authors for the response and clarification. I stand by my score in the light of it.

**Experience Assessment:**

I have published in this field for several years.

**Review Assessment: Checking Correctness Of Derivations And Theory:**

N/A

**Review Assessment: Checking Correctness Of Experiments:**

I carefully checked the experiments.

**Review Assessment: Thoroughness In Paper Reading:**

I read the paper thoroughly.

---

> ### Author Response · Authors · 2019-11-14
> **Thank you for your comment!**
>
> Thank you for your fruitful comment! Please find our response to your question below.
>
> >> One main issue that prevents this reviewer from increasing the rating is the measure of speed the authors use: counting exit at every token. This is a fine measure for inference time, but during training the model proceeds on the whole sequence, right? Could you provide speed numbers for the training step? Is there any improvement over a baseline Transformer or is this technique solely for inference?
>
> The focus of adaptive depth is to improve inference speed. Training speed is actually slightly decreased compared to a standard Transformer because we need to compute N output classifiers instead of just a single output classifier, however, since training benefits heavily from batching, the overhead of this is very minimal. On the WMT En-Fr benchmark we only see a marginal slowdown of 1% when training multiple output classifiers compared to just one for the baseline. We clarified this in the update to the paper we just posted.

---

### Official Review · AnonReviewer1 · 2019-10-23
**Official Blind Review #1**

**Rating:** 6

**Review:**

This paper studies using dynamic computation to alter the number of Transformer decoding layers each token uses to translate a given sentence. The paper considers using two variants of losses:  aligned training - same layer wise prediction loss for all tokens, and mixed training: loss on the output of a random layer for each token. For sampling the different layers the paper considers different distributions based on likelihood and the prediction rate.

The paper experiments these different training strategies on IWLST and WMT datasets. Training with token level sampling based on likelihood results in models that have smaller average exit (number of layers used in prediction) while preserving the BLEU scores of the standard Transformer training. Overall I believe the problem considered is interesting and the paper did a good job in setting up the problem and explaining the experimental setup results.

The paper mainly needs to improve in discussing existing work. Universal transformers also study dynamic computation based on input tokens. While the training setup is different here, without the large shared Transformer layers, and this paper mainly focuses on the dynamic halting strategies, it is important to discuss these differences in detail in the paper.

Introducing a classifier in each layer (W_n) increases the number of parameters and compute by N. How do you handle this?

The exit loss (eqn 3) is a cross entropy loss, which will have trouble when q^* and q have different supports. Doesn’t a dirac delta q^* cause problems here?


Minor:

Intro: first line of second paragraph needs to be rewritten.

There is a conflict in writing in both abstract and introduction as some sentences say that current models use the same computation irrespective of hardness of the input, followed by discussion of Universal Transformers, which do adaptive computation based on input hardness. The writing flow needs to be fixed in both abstract and intro.

** Post response update **

After reading other reviews I think my initial rating 8 might have been a bit higher and am adjusting to 6. Fixing the discussion about related works , as other reviewers also mentioned, will improve the paper.


**Experience Assessment:**

I have published one or two papers in this area.

**Review Assessment: Checking Correctness Of Derivations And Theory:**

N/A

**Review Assessment: Checking Correctness Of Experiments:**

I assessed the sensibility of the experiments.

**Review Assessment: Thoroughness In Paper Reading:**

I read the paper at least twice and used my best judgement in assessing the paper.

---

> ### Author Response · Authors · 2019-11-14
> **Thank you for your comments!**
>
> Thank you for your fruitful comments! Please find our response to your questions below.
>
> >> The paper mainly needs to improve in discussing existing work. Universal transformers also study dynamic computation based on input tokens.
>
> There are a number of differences to universal transformer (UT): UT repeatedly applies the same layer for a number of steps, while as our approach applies different layers at every step. The dynamic computation in UT considers a single type of mechanism to estimate the number of steps (=network depth) while as we consider a number of mechanisms and supervision regimes. Moreover, dynamic computation was not actually used for the machine translation experiments in the UT paper (it was used for other tasks though), the authors used a fixed number of steps for every input instead.
>
> We emphasized these differences in our updated paper.
>
>
> >> Introducing a classifier in each layer (W_n) increases the number of parameters and compute by N. How do you handle this?
>
> The number of parameters are only increased if we do not share the weights of output classifiers between each layer. We found a slight benefit when sharing output classifier weights for WMT En-Fr but not on IWSLT De-En.
> At inference time, only a single classifier is ever used (except for confidence thresholding), which does not change compute (in fact, we lower it when we use fewer layers).
> During training, compute requirements only increase very slightly when training N classifiers. On WMT En-Fr training time increased only by about 1%, keeping everything else equal. We added this to the updated paper.
>
> >> The exit loss (eqn 3) is a cross entropy loss, which will have trouble when $q^*$ and q have different supports. Doesn’t a dirac delta $q^*$ cause problems here?
>
> This is a good point. Since the distributions are discrete, the right delta is the Kronecker delta with 1 on the exit chosen by the oracle and 0 otherwise. The support of q and $q^*$ is within {1,..,N}
> so $H(q^*, q) = - \sum_{n=1}^N q^*(n) \log q(n)$
>
>
>
> >> … some sentences say that current models use the same computation irrespective of hardness of the input, followed by discussion of Universal Transformers, which do adaptive computation based on input hardness.
>
> We will make this more clear. With current models, we refer to state-of-the-art architectures in NMT. Universal transformers do not report results on machine translation with dynamic computation, instead, they use a fixed number of steps regardless of the input - this is not dynamic computation. We clarified this in the updated version of the paper.

---

### Author Response · Authors · 2019-11-14
**Paper updates**

We just updated the paper based on the suggestions of the reviewers. This includes:
* More thorough discussion of related work, particularly, differences to Universal Transformers (Introduction).
* Added training time comparison between standard Transformer and Aligned model (Section 4.2)
* Appendix A, added Algorithm 1: pseudo code for gradient scaling.
* Added Appendix B: Details how we tuned the confidence thresholding.
* Added Appendix C: ablation study of the depth regularization and the kernel smoothing.

---

### Author Response · Authors · 2019-12-20
**Paper updates #2**

We thank the reviewers and the PC for their feedback!
As promised, we updated the paper to address the two remaining concerns (issues #7 and #8) of R3:

1) We significantly improved results by tuning the exit thresholds for the binomial classifiers $\chi_t^n$ (on a held out set). Previously, we simply used 0.5 as a threshold. This improved results compared to the aligned baseline by a considerable margin on both benchmarks.

2) We added more results on the large scale benchmark (WMT'14 En-Fr) with the likelihood and correctness based oracles at the token level. We use the same training scheme as for IWSLT'14 De-En i.e we keep fine-tuning the encoder-decoder in the second phase of training.

With these changes we reduced the number of required decoder layers compared to the baseline for IWLT'14 De-En to 25% and that of the much larger  WMT'14 En-Fr to 40%.

---

### Decision · Program_Chairs · 2019-12-19

**Decision:**

Accept (Poster)

**Comment:**

This paper presents an adaptive computation time method for reducing the average-case inference time of a transformer sequence-to-sequence model.

The reviewers reached a rough consensus: This paper makes a proposes a novel method for an important problem, and offers reasonably compelling evidence for that method. However, the experiments aren't *quite* sufficient to isolate the cause of the observed improvements, and the discussion of related work could be clearer.

I acknowledge that this paper is borderline (and thank R3 for an extremely thorough discussion, both in public and privately), but I lean toward acceptance: The paper doesn't have any fatal flaws, and it brings some fresh ideas to an area where further work would be valuable.